# Smart Growth and Smart Shrinkage: A Comparative Review for Advancing Urban Sustainability

Yang Yang [1,2], Zhe Dong [1], Bing-Bing Zhou [1,*] and Yang Liu [1]

1   School of International Affairs and Public Administration, Ocean University of China, Qingdao 266100, China; yangyang@ouc.edu.cn (Y.Y.); zhe1997@stu.ouc.edu.cn (Z.D.); lyiu0917@stu.ouc.edu.cn (Y.L.)
2   Institute of Marine Development, Ocean University of China, Qingdao 266100, China
*   Correspondence: bbz@ouc.edu.cn

**Abstract:** In the context of ongoing global urbanization, the disparity in urban development, marked by the dual phenomena of urban sprawl and urban shrinkage at the regional level, has become increasingly evident. In this vein, two land-related governance strategies—smart growth (SG) and smart shrinkage (SS)—emerge as potential remedies to these challenges, targeting urban expansion and shrinkage, respectively. This study bridges the gap in the fragmented discourse surrounding SG and SS by conducting a comprehensive comparative review on the respective literatures. Utilizing the Scopus database, our research employs trend analysis, text and topic mining, time node analysis, and regional analysis, augmented by qualitative reviews of seminal papers. The findings reveal a notable shift in research focus, with interest in SS surging around 2010 (the number of SS-related papers published after 2010 accounts for 92.3% of the total number of the entire study period) as attention to SG waned, suggesting an impending paradigm shift in urban sustainability. The analysis indicates that SS research lacks the disciplinary diversity, thematic breadth, and empirical depth of SG studies, underscoring a need for a more robust theoretical foundation to support urban sustainability. Furthermore, while both SG and SS derive from environmental science foundations, SG predominantly addresses the physical and landscape attributes of urban areas, whereas SS focuses more on socio-economic dimensions. Our findings point to an intrinsic link between SG and SS, which could lay the groundwork for their integration into a unified theoretical framework to better advance urban sustainability.

**Keywords:** smart growth; smart shrinkage; urban sustainability; spatial governance; comparative review





## 1. Introduction

As the global landscape continues to urbanize, the question of how to navigate this transformation in a manner that advances sustainability has become paramount. This urgency is rooted in the recognition that urbanization is not just a demographic shift but a complex, multifaceted phenomenon with profound implications for sustainability [1–4]. According to the United Nations' forecasts in 2019, 56.15% of the world's population resides in urban areas, a figure that is expected to further rise to 67% by the middle of this century and reach 85% by the end of this century, signifying an unprecedented concentration of humanity in urban spaces [5,6].

However, this long-term global trend of urbanization is accompanied by simultaneous urban shrinkage in some regions during some periods [7]. On the one hand, urban growth dominates most parts of the world, leading to urban sprawl in some areas that is characterized by dispersed, low-density expansion of urban land use [8,9]. Urban sprawl leads to a suite of problems such as farmland depletion, environmental degradation, and the financial burden of infrastructure investments. On the other hand, amid long-term global urbanization, shrinking cities emerged in Germany after the collapse of the Berlin Wall

and have occurred in various parts of the world [10], leading to intertwined issues such as spatial decay, economic downturn, and rising crime rates [11]. Generally speaking, urban growth and urban shrinkage are now present as two parallel research discourses in the urbanization-related literature, each targeting distinct challenges and opportunities [12,13].

In response to urban sprawl and shrinkage, two respective paradigms emerged in the academic and professional communities, often known as smart growth (SG) and smart shrinkage (SS). On the one hand, SG can be interpreted as the third wave of land planning rooted in roughly the late 1990s, following growth controls and subsequent comprehensive land planning [14]. The targeting issues of SG are more than mitigating environmental degradation by containing urban sprawl and coordinating infrastructure provision by comprehensive land planning, and also include urban revitalization and placemaking, thus demanding the coordinated planning of multiple land uses and associated spatial governance policies. In this vein, the "smartness" of SG reflects the demarcation of prioritized urban development areas with a package of advantageous policies like tax incentives, grants, and infrastructure investments. At the core of SG are compact urban development and mixed use of urban land, with comprehensive considerations of related issues crucial to urban vitality, such as housing, transportation, land use, infrastructure, urban design, and environmental sanitation [14–17]. In contrast, SS is a relatively new term and one of quite a few similar terms promoting the philosophy of treating cities from no longer a growth-oriented perspective but focusing instead on reduced urban population and associated problems [18]. Popper and Popper first coined "smart decline" to advocate planning for fewer people, fewer buildings, and less land development [19]. Later, the establishment of an EU research network on urban shrinkage, SHRINK SMART, helped popularize the term of SS, with "smart" strategies like land adjustment, land banking, relocation assistance, and urban renewal, although similar terms like "smart decline" and "right-sizing" are still in use [18]. Mirroring the largely parallel literatures on urban growth and shrinkage, the "smartnesses" of SG and SS remain largely unintegrated [13], falling short of their potential in better advancing urban sustainability.

Increasingly, more and more studies have started to recognize the coupled landscape of urban growth and shrinkage at regional and global scales [20,21]. Especially at national and subnational scales, interregional linkages such as economic competition and population migration among various social-environmental systems make up a metacoupled polycentric system of development and residence. From a core–periphery systems perspective [22–24], the growth of a core city might well be linked to the shrinkage of its relatively peripheral areas. In this sense, the full spectrum of urban development challenges should be widely recognized, the metacoupled and evolving nature of urban growth and shrinkage deserves explicit acknowledgement, and further, the "smart" wisdom of SG and SS merits better integration for advancing global and regional urban sustainability, for which timely research efforts are needed.

To better unleash the potential of SG and SS in advancing urban sustainability, our research seeks to address the following three questions: (1) What are the bibliometric statistical characteristics of SG- and SS-related literature? (2) What are the differences between and similarities of SG and SS? (3) How can SG and SS to integrated to better support sustainable urban development? To address these three questions, following the research design and methodological approach in Zhou et al. [25], we present a multidimensional comparative analysis of SG and SS in terms of research trends, thematic evolution, disciplinary structure, global distribution of research areas, and knowledge base. The remaining of the paper is structured as follows: Section 2 outlines the methodology, including data sources and analytical techniques. Section 3 presents comparative findings of SG and SS. Section 4 highlights the caveats for interpreting results and generalizing findings, clarifies the two research domains' conceptual linkages and differences, and proposes directions for integrating SG and SS into a cohesive theoretical framework that can maximize the potential of their "smartnesses" in advancing urban sustainability. Section 5 concludes the paper with key messages.

## 2. Materials and Methods

Our entire research procedure was mainly divided into three stages (Figure 1). Firstly, we selected a suitable literature database based on the research needs, established literature screening criteria, and conducted literature retrieval and collection. Secondly, we conducted a systematic comparative analysis of SG and SS based on the bibliographic data of the collected papers, including publication and citation trend analyses, topical and thematic analyses, disciplinary structure, and global coverage. Finally, we interpreted the content of the top 10 most-cited publications in SG and SS respectively, in order to cross-validate the conclusions drawn from the data analysis.

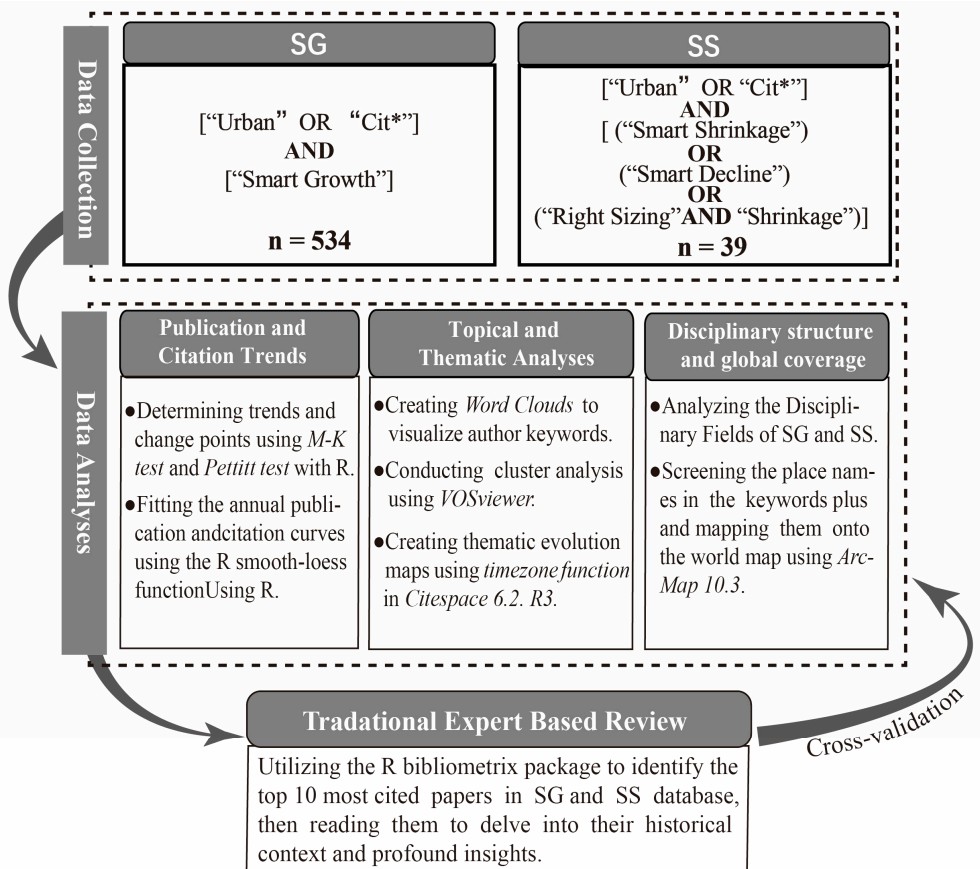

**Figure 1.** Flowchart of data collection and analyses. The symbol * serves as a wildcard, representing any sequence of characters or suffixes. For instance, "cit*" includes words such as "cities" or "city".

### 2.1. Data Collection

Referring to the literature of Zhou, Wu, and Anderies [25], we sourced the data from the Scopus database, as of 24 March 2023. Recognized as the most extensive abstract and citation database globally, Scopus boasts broader literature coverage, disciplinary range, and global journal representation across both Northern and Southern Hemispheres, outperforming the Web of Science in these aspects [26]. In addition, Scopus established the International Center for the Study of Research in 2019, composed of bibliometrics experts [27]. This ensures Scopus's high usability in the field of bibliometrics, and as a result, it is extensively used in bibliometric studies across various disciplines [28,29].

Regarding the literature search, direct usage of the term "Smart Growth" as the sole keyword resulted in retrieving a significant number of irrelevant publications outside the urban studies field. To mitigate this, the SG search was narrowed to urban research specifically by employing the terms "Cit*" or "Urban" in conjunction with "Smart Growth," forming the query ["Cit*" OR "Urban"] AND ["Smart Growth"]. Additionally, considering the diverse nomenclature for SS, including terms like Smart Shrinkage, Smart Decline, and

Right Sizing, a comprehensive search query ["Urban" OR "Cit*"] AND [ ("Smart Shrink*") OR ("Smart Decline") OR ("Right Sizing" AND "Shrink*")] was formulated to encompass the varied terminology associated with SS. The execution of these search strategies was further refined by restricting the publication date to include only articles published up to 2022, and specifying the document types to "Article" and "Review". Considering that English is the most widely used language in the world and also the universal language for international communication in academic fields, we limited the language of search results to English. In the end, a collection of 534 papers on SG and 39 papers on SS was sampled. The bibliographic details of these papers were extracted and served as the foundation for the subsequent bibliometric analysis (Table 1).

**Table 1.** A comparative overview of sampled papers on SG and SS.

|  | SG Subset | SS Subset | Ratio of SS to SG |
| --- | --- | --- | --- |
| Number of articles | 534 | 39 | 7.3% |
| Timespan | 1997–2022 | 2002–2022 | 80.77% |
| Number of authors | 1078 | 80 | 7.42% |
| Number of sources | 224 | 25 | 11.16% |
| Annual growth rate | 10.8% | 7.18% | 66.48% |
| Number of Keywords Plus | 1685 | 133 | 7.89% |
| Number of author's keywords | 1127 | 105 | 9.3% |

### 2.2. Publication and Citation Trend Detection

To explore the evolution and identify potential inflection points within the research domains of SG and SS, this study employed a time series analysis on yearly publication and citation data. The analysis utilized the Mann–Kendall test (a non-parametric test used to identify trends in time series data) and Pettitt's test (a non-parametric test used to detect change points in time series data), as implemented in the R "trend" package, to assess trends and changes in publication and citation frequencies over time. For visual representation of these trends, the R "ggplot2" package was utilized, specifically leveraging the "loess" method within the "geom_smooth" function (a function for performing smoothing operations in R, used to reduce noise or abrupt changes, thereby revealing underlying patterns or trends within the data) to model and depict the annual variations in publication and citation counts [30]. Furthermore, to examine the relationship between annual publication and citation counts for SG and SS, the correlation function from the R "see" package was applied. The results of this analysis were then illustrated through a correlation heatmap, providing a visual representation of the interconnectivity between publication output and citation impact in these fields.

### 2.3. Text Mining and Thematic Mapping

The initial step involved recording the quantity of publications within each disciplinary field related to SG and SS as categorized by Scopus. This was followed by employing Excel and Adobe Illustrator to graphically represent the disciplinary spread of these research areas. Subsequent bibliometric analysis was performed using the R Bibliometrix package, which facilitated the identification and consolidation of key terms, the elimination of irrelevant data, and the creation of a keyword-based word cloud via www.wordclouds.com (accessed on 15 April 2023) [25,31,32]. Additionally, VOSviewer was utilized to construct and visualize keyword co-occurrence networks for both SG and SS, allowing for the deduction of primary research themes from the interlinking patterns of keywords [33]. The study also leveraged the timezone function in Citespace 6.2. R3 to generate a timeline graph, marking the chronological emergence of significant terms and thus tracing the historical development of topics within SG and SS research [34]. Moreover, a detailed examination of keywords led to the identification and categorization of geographical names, which were then mapped globally using ArcMap 10.3 to display the spatial concentration and

thematic focus of the research locales across various regional levels, offering insights into the geographic dispersion and focal points of studies within the SG and SS frameworks.

*2.4. Content Analysis*

In addition to the above-noted quantitative assessments, we utilized the R Bibliometrix package to pinpoint the top 10 most-cited articles within both the SG and SS literature groups, resulting in a total of 20 key papers. This approach identified papers that were frequently cited within their respective groups, indicating their foundational importance to the field. These pivotal articles were then meticulously reviewed to ensure the bibliometric findings' accuracy and to gain more profound insights into the research directions and developmental trajectory of SG and SS. Such a quantitative–qualitative combined thorough examination [25] enabled us to deepen our understanding of the core themes and progress within these research domains.

## 3. Results

*3.1. Publication and Citation Trends*

SG-related research emerged in 1997, and then increased rapidly in terms of publication number over a span of 26 years until 2022 (Figure 2a), with its annual growth rate averaging 10.8% (Table 1). Notably, the growth rate started to slow down around 2006 and it peaked in 2013 (30 papers), followed by a gradual decline. A change point in SG publications occurred around 2006, which is statistically significant. Overall, SG research activity showed a trend of rapid outbreak, stable growth, and slow decline. In terms of the trend in annual citation counts, SG papers published around 2005 received the highest number of citations (Figure 2b).

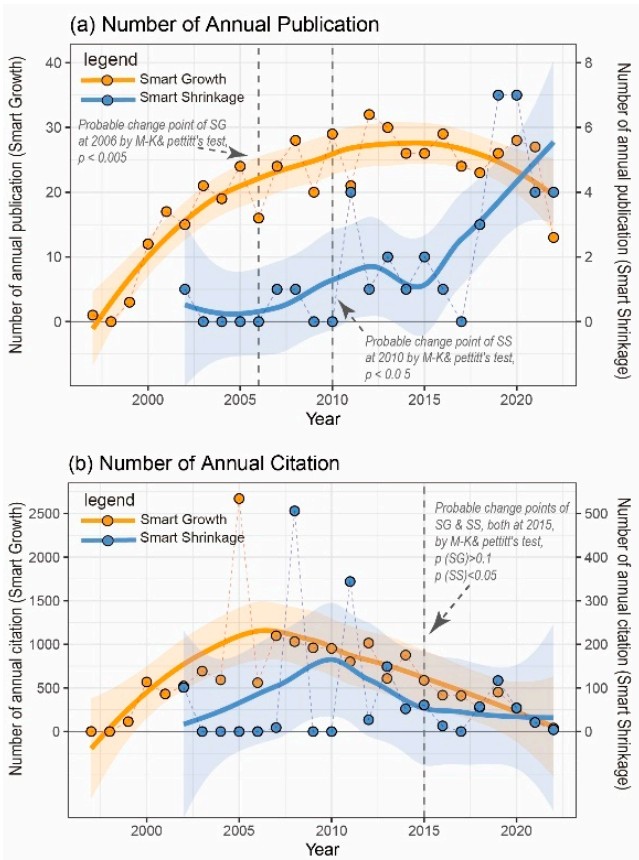

**Figure 2.** Correlation and trends in annual publications and citations within SG and SS research. (**a**) Yearly publication trends in SG and SS. (**b**) Yearly citation trends per paper in SG and SS datasets. The shaded area illustrates the 95% confidence interval of the smoothing function.

SS-related research began in 2002, and over a span of 21 years until 2022, its publication count grew annually by 7.3% (Table 1). From 2002 to 2010, SS research went through a period of tepid and prolonged incubation, with 6 out of 9 years having no publications (Figure 2a). Notably, a statistically significant change point in SS publication trend occurred in 2010. There was a minor fluctuation in its annual publication counts during 2010-2015, followed which the past few years saw dramatic growth of SS research output. In terms of the trend in annual citation counts, SS papers published around 2010 received the highest number of citations (Figure 2b). From a comparative perspective, SS papers appeared 5 years later than SG, and SG had over 10 times the quantity of SS in terms of the number of source journals, authors, Keywords Plus, and author's keywords. It is safe to say that SS research intensity is much lower than that of SG. However, a trend seems to be emerging: SG research intensity has started to decline in recent years, while SS research intensity keeps rapidly increasing. Additionally, SG and SS studies present similar research impacts as indicated by annual citation counts: they both show an initial increase followed by a decrease, although with SG reaching its peak earlier than SS.

### 3.2. Multi-Faceted Research Topics and Themes

#### 3.2.1. Prominent Author Keywords

Prominent author keywords reflect scholars' conceptual bases of their SG and SS studies (Figure 3). The two research domains have 33 shared author keywords, mainly focusing on urban planning, landscape coverage, and climate environment. Such shared author keywords account for about one-third of the total author keywords for SS and one-thirty-fourth of those for SG. Among them, "Right-sizing", "Urban Shrinkage", "Vacant Land", and "China" appear more than twice in the SS database, while "Urban Planning", "Sustainable Development", "Climate Change", "GIS", "Regional Planning", "Quality of Life", and "Regionalism" appear more than five times in SG papers.

(a) Smart Growth

(b) Smart Shrinkage

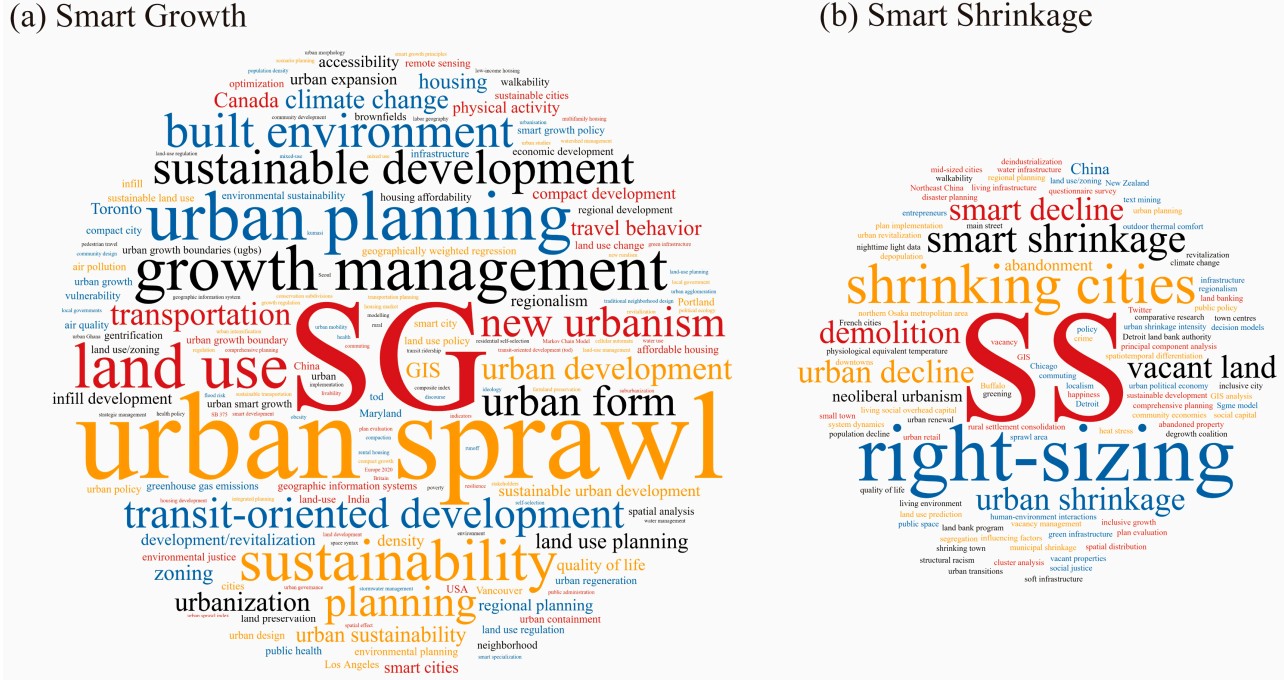

**Figure 3.** Visual representation of author keywords in SG and SS publications. (**a**) Frequently occurring SG keywords. (**b**) All SS keywords. Font size indicates keyword frequency, with comparative weights set for SG and SS for balanced analysis.

The difference lies in the fact that SG's frequently used keywords are much more diverse compared to those in SS. Compared to 1119 effective author keywords in SG (an average of 2.10 author keywords per paper), SS only has 100 effective author keywords

(an average of 2.56 effective author keywords per paper). A total of 97% of the author keywords in SG are not present in SS, and besides topics related to urban growth itself, frequently used keywords with high frequencies include "Sustainability" (21 times), "Built Environment" (17 times), "New Urbanism" (15 times), "Transit-oriented Development" (15 times), and "Urban Form" (15 times). In general, the unique frequently used keywords in SG compared to SS are related to classical theories on one hand and lean towards practical urban planning on the other. On the contrary, the frequently used keywords in SS focus on the phenomenon of urban shrinkage and the concept of SS itself (Figure 3), indicating that SS is still in the stage of conceptual construction and theoretical research.

Notably, "Sustainable Development" appears 16 times in SG, while it only appears once in SS; "Sustainability" appears 21 times in SG but is not present in the author keywords of SS. Although the concepts in SS align with sustainability science, SS focuses more on specific issues in the urban shrinkage process, while the forward-looking and goal-oriented perspective emphasized by sustainability science [35] receives little attention in SS.

### 3.2.2. Main Research Themes

To reveal the thematic structure of sampled SG and SS papers, the keyword co-occurrence networks of Keywords Plus of the two research domains were visualized [25].

For SG, its 49 Keywords Plus that occur 15 or more times fall into four main themes (Figure 4a): the red cluster emphasizes land use, urban areas, environment, humanities, and the United States (*n* = 14), mainly reflecting urban landscapes; the green cluster emphasizes planning, mostly related to management measures and policies for urban growth (*n* = 13), reflecting management strategies; the blue cluster emphasizes SG and keywords related to urban development concepts (*n* = 11), mainly reflecting theories and concepts; and the yellow cluster emphasizes China and issues related to urban expansion, urban growth, GIS, and spatial analysis methods (*n* = 11), mainly reflecting empirical research. The morphology of the co-occurrence network, featured by the cohesiveness within each theme, indicates the SG research is relatively mature with four recognizable subfields.

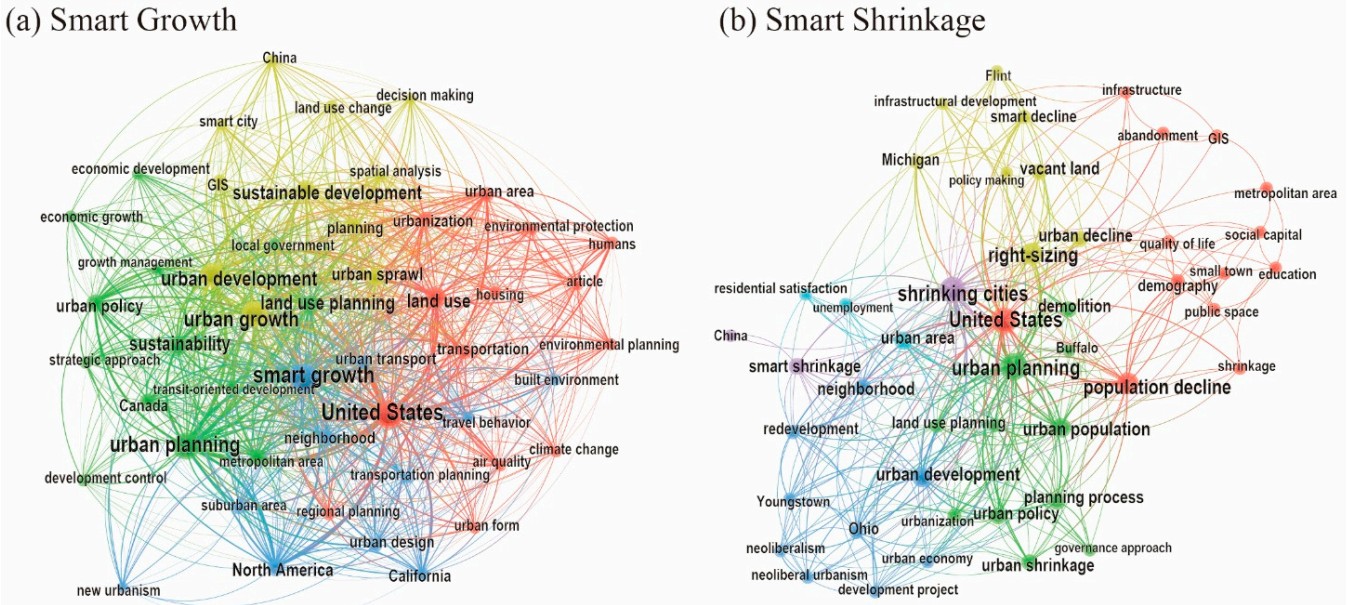

**Figure 4.** Network analysis of keyword co-occurrence in SG and SS. (**a**) SG with keyword frequencies ≥ 15. (**b**) SS with keyword frequencies ≥ 2. Frequency thresholds vary to align the networks' keyword density across SG and SS.

For SS, its 47 Keywords Plus that occur two or more times group into six themes (Figure 4b). There are no clear boundaries between these six clusters, and the keywords within each cluster are loosely connected; thus, it is difficult to synthesize their shared

theme. This morphology of the SS Keyword Plus co-occurrence network indicates that research related to SS is still in its formative stages.

### 3.2.3. Thematic Evolution

To further reveal the temporal evolution of research themes and the emergence of new topics [34] in the two research domains, thematic evolution maps were produced to display the years in which author keywords first appeared in the SG and SS paper datasets. On the SG side (Figure 5), its number and frequency of new keywords have decreased over time. Even when the threshold for mapping frequent keywords was reduced from five times to two times, the decreasing trend of new keyword emergence remained consistent. In particular, no keywords have occurred five or more times since 2018, with "smart city" being the last keyword occurring more than five times in 2017. Notably, the chronological appearance of SG keywords clearly shows that SG originated from "growth management" in relation to "urban growth" and "land use planning" as well as "urban planning", and that the research discourse has evolved from SG, to "smart city", and increasingly toward "sustainable city"—although "sustainable development" and "sustainability" had become key terms in the SG literature dating way back to the early 2000s.

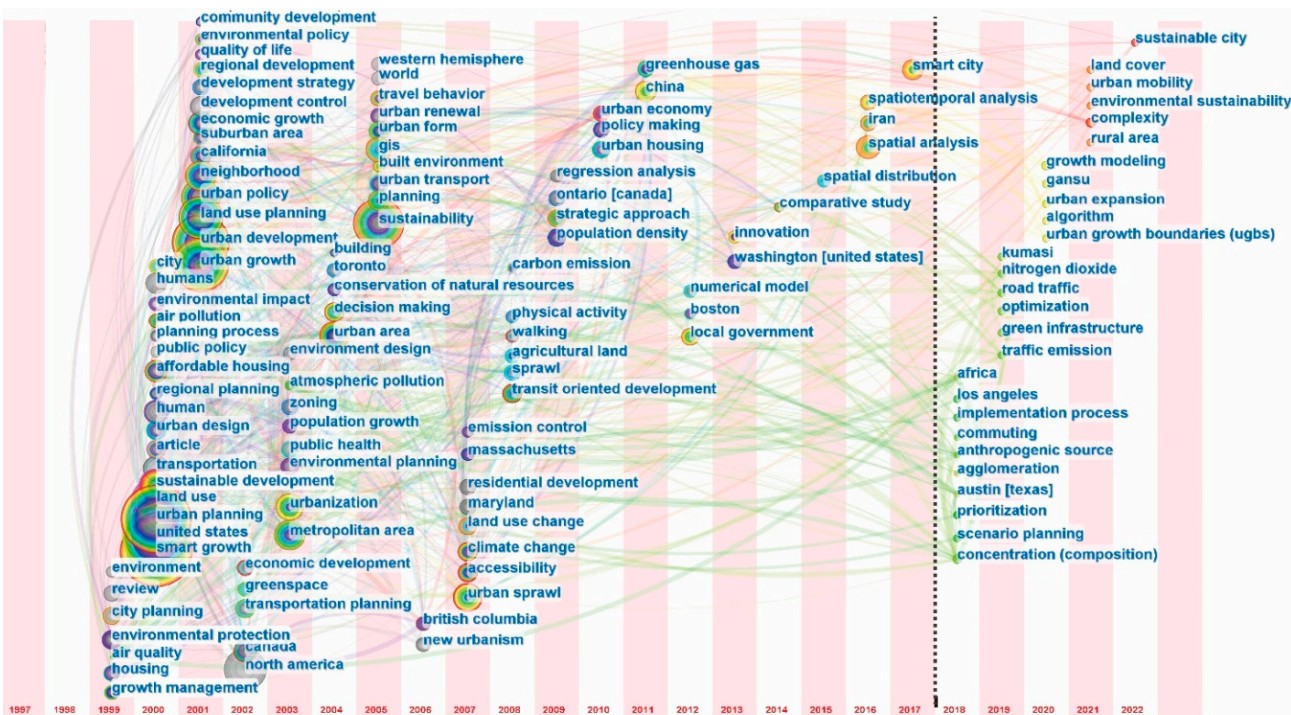

**Figure 5.** Chronological appearance of keywords in SG research. Created using *Citespace* 6.3.R2, showcasing years of first occurrence and frequency of keyword appearances. The gap in 1997 is noted due to the absence of keywords from Tregoning (1997) [15].

SS is opposite from SG in terms of its overall increasing trend of new keywords' number and frequency (Figure 6). Clearly, SS originated from "land use planning" and "urban planning" as early as 2002, although scholarly interest in researching SG did not regain momentum until 2008 when concerns on "shrinking city" and "right sizing" became elevated. Later in 2011, "smart decline", a term similar to SG, gained popularity, while SS per se did not become a popular term until 2018. Notably, the SS studies before 2018 featured popular terms such as "urban development", "development project", "redevelopment", "residential satisfaction", and "quality of life", indicating more of a growth-oriented, shrinkage-mitigation thinking, whereas SS studies after 2018 present growing concerns on "rightsizing", "small town", and "degrowth coalition", suggesting a shrinkage-adaptation thinking. Another interesting feature of SS studies after 2018 is their attention to "con-

ceptual framework", "governance approach", various methodological issues, and diverse study areas. A more subtle yet important observation is that "sustainable development" and "sustainability" have never been popular terms in the SS literature.

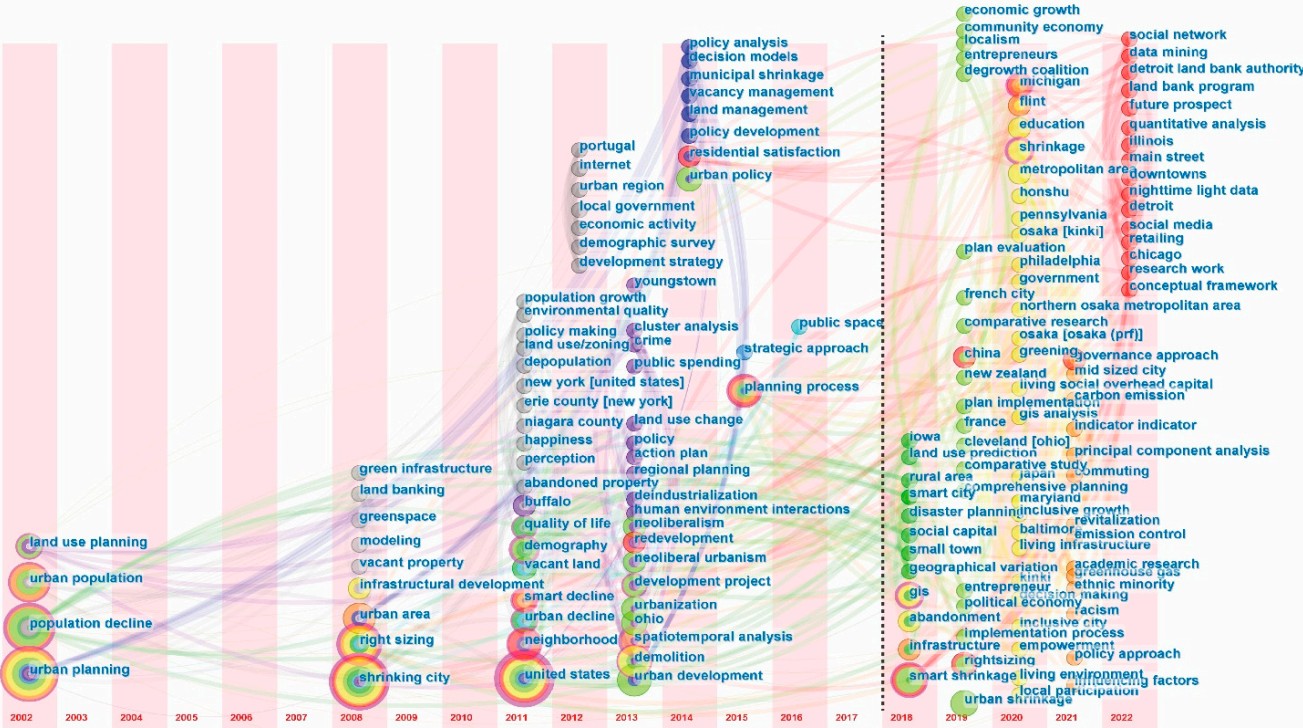

**Figure 6.** Evolution of SS research keywords over time. Displays keyword frequencies ≥2 up to 2017 and ≥1 from 2018 onwards. Other methodological settings are consistent with those employed in Figure 5.

### 3.3. Disciplinary Structure and Global Coverage

#### 3.3.1. Contributing Disciplinary Fields

Based on the classification of disciplinary fields in Scopus, the sampled SG papers fall into 18 fields, encompassing the 10 fields that those SS papers belong to (Figure 7). In this sense, SG involves more diverse disciplinary perspectives than SS. Furthermore, in both research domains, the category of social sciences holds an absolute majority (78.09%), followed by environmental science (37.67%). Similarly, 38 out of the 39 SS papers belong to social sciences (97.44%), followed also by environmental science (30.77%). Such similarity reflects the dominance of environmental social science in researching SG and SS. Additionally, the proportion of engineering papers in SG (12.92%) is significantly higher than in SS (5.13%), indicating the slightly greater involvement of natural sciences in researching SG than SS, as exemplified by engineering.

#### 3.3.2. Global Coverage of Study Areas

The study areas covered by SG are significantly broader than those covered by SS, at national, provincial/state, and city scales (Figure 8). SG includes a total of 182 place name keywords, accounting for approximately 10.8% of the total SG Keywords Plus, encompassing 37 countries, 46 provincial/state-level administrative regions, 99 cities, and other locations. SS includes a total of 24 place name keywords, accounting for 18% of the total SS Keywords Plus, including 8 countries, 5 provincial/state-level administrative units, and 9 cities and other locations.

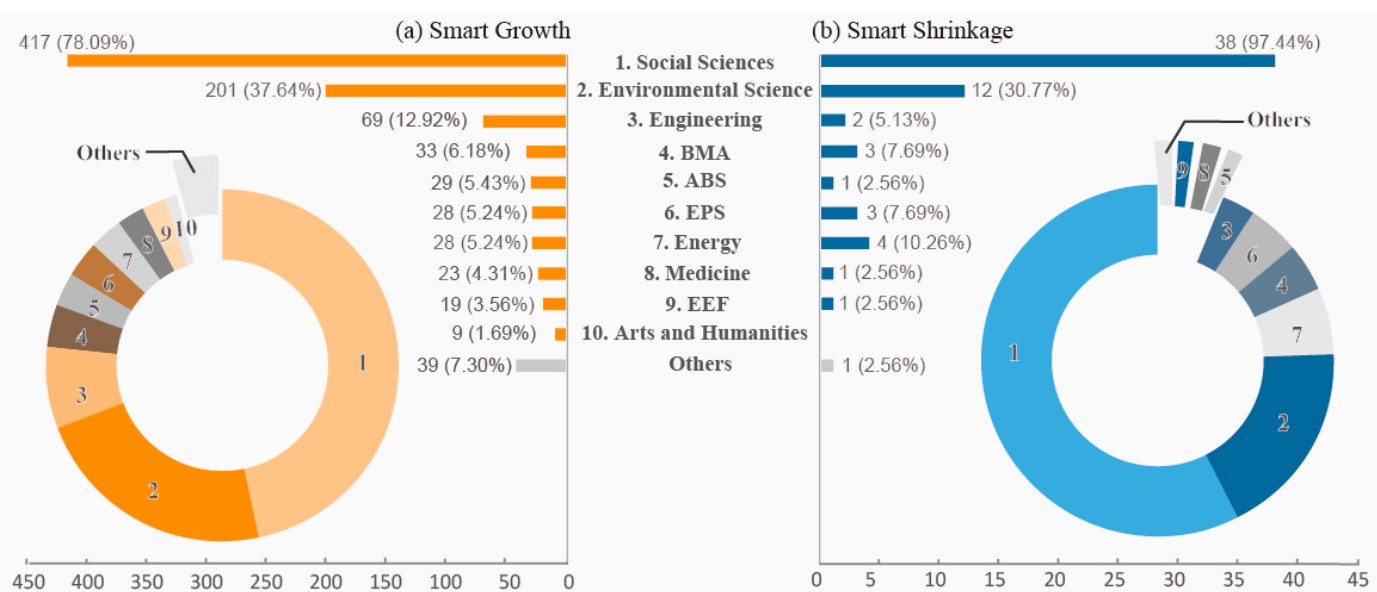

**Figure 7.** Comparative analysis of disciplinary fields in SG and SS research. (**a**) Distribution in SG. (**b**) Distribution in SS. Bars indicate paper counts with percentages showing field proportions within each dataset. Pie charts reflect field distribution against the total. Notes: (1) Papers may span multiple disciplinary fields; 67.23% of SG and 69.23% of SS papers fall into this category. (2) SG's x-axis scale is tenfold that of SS. (3) Disciplinary abbreviations, such as BMA, ABS, EPS, and EEF are used for clarity, denoting "Business, Management and Accounting", "Agricultural and Biological Sciences", "Earth and Planetary Sciences", and "Economics, Econometrics and Finance", respectively. The "Others" in this graph may be classified under less-fitting disciplines by the Scopus algorithm due to some words in the abstracts. However, their number is small, and their main content does not deviate from the research themes of SG and SS.

At the national scale, the study areas of high interest in SG are the United States (mentioned 195 times), followed by Canada, China, various European countries, and Iran. They also involve numerous economically underdeveloped countries in Asia and Africa, illuminating all continents except Antarctica. Similarly, the most mentioned country in SS is the United States (mentioned 14 times), with other countries like Mexico, Japan, South Korea, France, and Portugal receiving much less attention. The countries involved in SS studies are relatively economically developed, with a knowledge gap remaining for economically underdeveloped regions in Africa and South America (the single point in South America is French Guiana, shown on the map as part of France as a whole) and vast areas of Asia beyond East Asia (white areas in Figure 8).

At the provincial/state scale, the distribution of provincial/state-level SG study areas is quite extensive, with over half of the states in the United States involved. The states/provinces that receive noticeable SG research attention are in China, Australia, Iran, and India. On the other hand, at the provincial/state scale, the study areas of SS are mainly in the Great Lakes region of the United States, including Ohio (three times), Michigan (three times), Pennsylvania (one time), Illinois (one time), Pennsylvania (one time), and Maryland (one time).

Similarly, at the city scale, the contrast in study area distribution is remarkable between SG and SS. In SG, the most attention is given to cities in North America, but areas with concentrated distributions of major cities in East Asia, Western Europe, and elsewhere are also covered. Highly studied cities in SG include Toronto (10 times), Washington (9 times), Seattle and Los Angeles (6 times each), and Austin, Boston, and Seoul (all 5 times). On the other hand, the focal cities of SS, apart from the Japanese city of Ōban [36,37], are in the Rust Belt region near the Great Lakes in the United States [38].

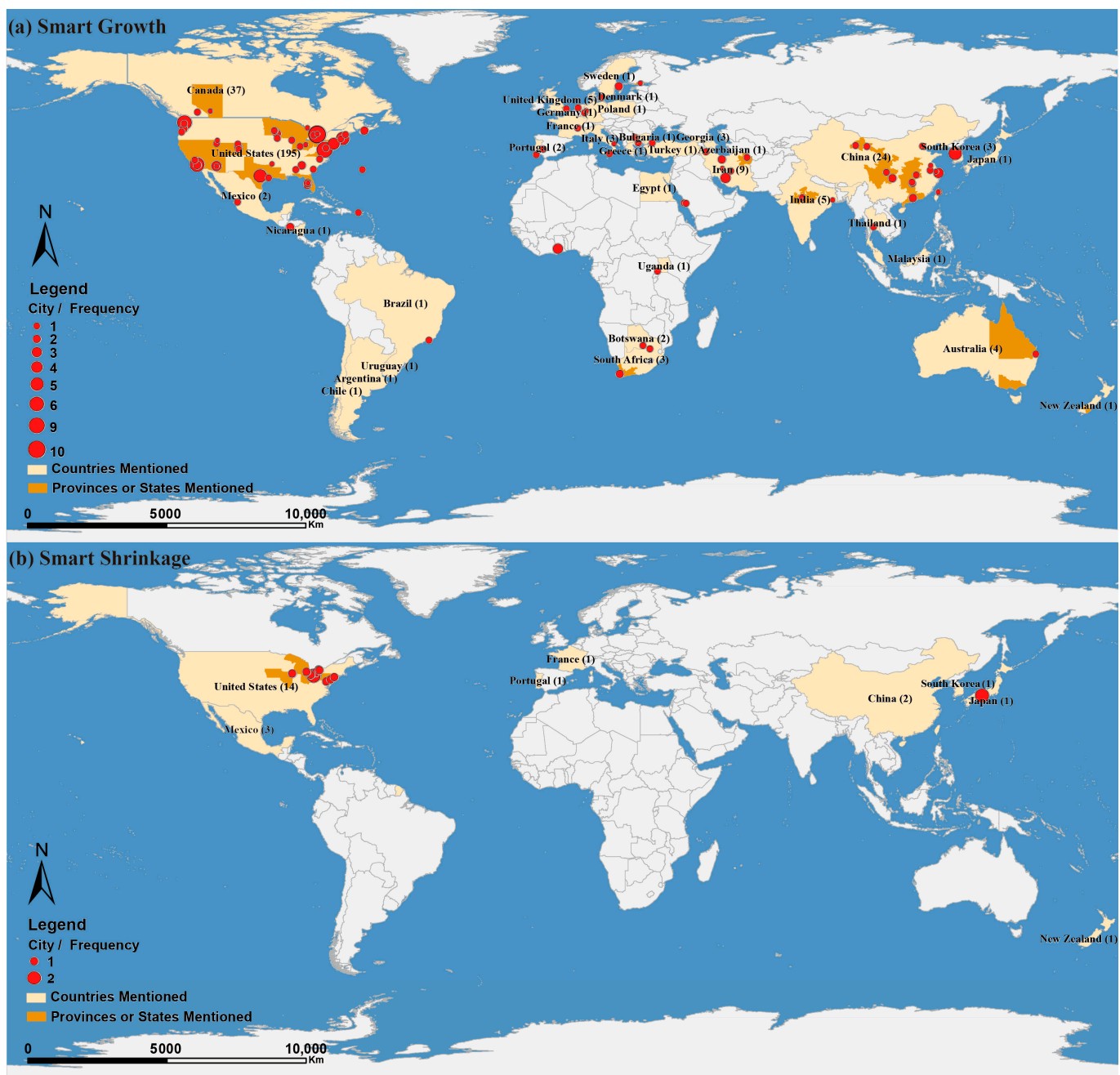

**Figure 8.** Geographical mapping of SG and SS research focus. Maps illustrate research distribution at national, provincial/state, and city levels, with data derived from Keywords Plus. Base map sourced from Natural Earth (does not represent the author's political views though), with missing place names geocoded via Google Maps for *ArcMap* 10.3 visualization.

## 3.4. Top Ten Most-Cited Publications of Two Research Domains

Complementary to analysis of contributing disciplinary fields, the most-cited references of a research domain indicate the knowledge base that has nurtured the referring papers. The top 10 most-cited papers in SG span from 1999 to 2007 (Table 2). The first four papers and the seventh paper discuss SG's conceptual and theoretical bases [39–43]. Around the time when the concept of SG emerged (around 1997), the American Planning Association, the Environmental Protection Agency, the Natural Resources Conservation Council, and the Maryland state government introduced policies and regulations related to SG management. At the same time, the Ground Transportation Project Department introduced the

"Smart Growth Toolkit." This top-down promotion quickly sparked widespread academic discussion and led to the formation of an academic community of smart growth—the Smart Growth Network. SG rapidly aroused research interest in the academic community during its emergence and its theoretical foundation was quickly developed in a series of planning practices. The ten SG principles promulgated by the Smart Growth Network have been widely recognized. The themes covered include spatial compactness, housing, transportation, community form, and open space, providing clear and detailed guidance for subsequent academic research and planning practice.

The other five papers are empirical studies. Among them, Song (2005) assessed the extent to which the land development patterns of five American counties conform to SG based on widely accepted SG principles, such as street network connectivity, density, land use mix, accessibility, and walkability [44]. The premise for conducting such assessments is that SG already has clear goals and principles to follow. The remaining articles examined SG in relation to more specific issues including housing [45], small communities [46], legislation [47], and transportation and land use [48]. These empirical focuses are long-lasting concerns of SG research and policymaking.

**Table 2.** Top ten most-cited publications of SG and SS.

| Sampled Papers | Most-Cited Publications | Title | Source | Cited Frequency |
|---|---|---|---|---|
| Smart Growth | Downs, 2005 [49] | Land preservation: An essential ingredient in smart growth | *Journal of Planning Literature* | 8.61% |
| | Downs, 2001 [40] | What does 'smart growth' really mean? | *Planning* | 3.93% |
| | Burchell et al., 2000 [41] | Smart growth: More than a ghost of urban policy past, less than a bold new horizon | *Housing Policy Debate* | 3.75% |
| | Knaap & Talen, 2005 [42] | New urbanism and smart growth: A few words from the academy | *International Regional Science Review* | 3.56% |
| | Song, 2005 [44] | Smart growth and urban development pattern: A comparative study | *International Regional Science Review* | 3.56% |
| | Danielsen et al., 1999 [45] | Retracting suburbia: Smart growth and the future of housing | *Housing Policy Debate* | 3.37% |
| | Daniels, 2001 [43] | Smart growth: A new American approach to regional planning | *Planning Practice and Research* | 3.18% |
| | Edwards & Haines, 2007 [46] | Evaluating smart growth: Implications for small communities | *Journal of Planning Education and Research* | 2.81% |
| | Talen & Knaap, 2003 [47] | Legalizing smart growth: An empirical study of land use regulation in Illinois | *Journal of Planning Education and Research* | 2.43% |
| | Handy, 2005 [48] | Smart growth and the transportation-land use connection: What does the research tell us? | *International Regional Science Review* | 2.06% |
| Smart Shrinkage | Schilling & Logan, 2008 [50] | Greening the rust belt: A green infrastructure model for right sizing America's shrinking cities | *Journal of the American Planning Association* | 56.41% |
| | Hollander & Németh, 2011 [51] | The bounds of smart decline: A foundational theory for planning shrinking cities | *Housing Policy Debate* | 41.03% |
| | Rhodes & Russo, 2013 [52] | Shrinking smart?: Urban redevelopment and shrinkage in Youngstown, Ohio | *Urban Geography* | 20.51% |
| | Hummel, 2015 [53] | Right-Sizing cities in the United States: Defining its strategies | *Journal of Urban Affairs* | 12.82% |
| | Popper & Popper, 2002 [19] | Small can be beautiful | *Planning* | 10.26% |
| | Hollander & Popper, 2007 [54] | Planning practice and the shrinking city: Reversing the land use allocation model | *Plan Canada* | 10.26% |

**Table 2.** *Cont.*

| Sampled Papers | Most-Cited Publications | Title | Source | Cited Frequency |
|---|---|---|---|---|
| Smart Shrinkage | Hollander, 2011 [55] | Can a city successfully shrink? evidence from survey data on neighborhood quality | *Urban Affairs Review* | 10.25% |
| | Hummel, 2015 [56] | Right-Sizing cities: A look at five cities | *Public Budgeting and Finance* | 10.25% |
| | Frazier et al., 2013 [57] | The spatio-temporal impacts of demolition land use policy and crime in a shrinking city | *Applied Geography* | 7.70% |
| | Hollander & Cahill, 2011 [58] | Confronting population decline in the Buffalo, New York, Region: A close reading of the Erie-Niagara framework for regional growth | *Journal of Architectural and Planning Research* | 5.13% |

The time span of the top 10 most-cited papers in SS ranges from 2002 to 2015. These papers indicate the ambiguity of SS concepts and theories. Popper (2002) proposed the idea of smart shrinkage, a precedent concept of SS, to advocate "planning for fewer people, fewer buildings, less land use" [19]. It is, however, more of a direction instead of actionable guidelines. SS as a revolution compared to traditional growth-oriented planning in the United States has been limited in planning practices, leading to much of the SS research remaining largely theoretical for a long period. During this time, filling and greening of vacant land emerged as an essential SS practice [50]. It was not until 2005 when the city of Youngstown published the renowned "2010 Comprehensive Plan" that SS truly began to be explored in planning practices [52]. Youngstown's planning was phenomenal at that time, and subsequently, cities like Flint, Cleveland, Detroit, and Buffalo also explored planning strategies to address urban shrinkage [56]. Along with exploration in SS planning practices, various policy tools began to emerge, and, in addition to measures such as demolition and reconstruction, land banking became an important approach to address the funding shortage for SS [53]. Notably, the study areas of these seminal SS-cited papers are largely limited to American cities such as Youngstown, Cleveland, Flint, Detroit, and Buffalo, pointing to a potential geographical bias in the knowledge base of SS.

## 4. Discussion

Urban development is a cyclic process from natural growth to natural decline and then to growth again. As cities around the world develop along different paths, urban sprawl and urban shrinkage often coexist [21,59]. For a long time, people have habitually treated the coexisting issues of urban sprawl and urban shrinkage in isolation. Correspondingly, the SG and SS strategies that have emerged in response to urban sprawl and urban shrinkage are rarely considered within the same context. This easily leads to a "either-or, this but not that" phenomenon in the process of sustainable urban governance. Therefore, based on the systematic comparative analysis of the literature on SG and SS, we analyzed the commonalities and unique characteristics of SG and SS, delineated the logical connection between the two research domains, and attempted to integrate them into a unified framework of "smart development".

### 4.1. Limitations

Before generalizing findings, three major caveats should be acknowledged. First, our sample size of the SS literature is relatively small, thus making our SS results sensitive to further publications (Figure 2a). Second, the sampled SS papers represent only research published in English, thus making our SS results biased toward English-speaking countries like the United States (e.g., Figures 4b and 8, Table 2). Third, the sampled SS papers explicitly mentioned in their title, abstract, and keywords the terms "smart shrink*", "smart decline", and/or "'right sizing' AND 'shrink*'" in the context of cities or urban areas, yet some studies on shrinking cities that use alternative terms like "urban redevelopment",

"urban revitalization", "urban regeneration", and/or "urban resurgence" are insufficiently reflected by our results [60].

Considering these caveats, our interpretation of SS results needs to be very careful to avoid overgeneralization. Essentially, the "smartness" captured in this study is only part of the wisdom that scholars have explored to address urban shrinkage (and urban sprawl); yet it is safe to claim that our results reflect how SS and SG, as two scholarly concepts, have been used in the English literature.

Nonetheless, it is imperative to acknowledge that our research, in its quest to explore the term "SS"—a term birthed in the United States—inadvertently neglects the manifold endeavors undertaken by Europe, situated on the other side of the Atlantic, to counter urban shrinkage. Indeed, the phenomenon of urban shrinkage surfaced earlier and is more pervasive in Europe, with European governments and academicians initiating actions ahead of their American counterparts [61]. In 2001, prior to the introduction of SS in the United States, Germany was grappling with the escalating issue of urban shrinkage in East Germany. In response, they proposed the Stadtumbau Ost (Urban Restructuring East) plan, which aimed at balancing the real estate market in shrinking cities through the demolition of vacant houses and the reinvigoration of urban allure. This initiative can arguably be considered humanity's inaugural attempt at a paradigm shift from growth planning to shrinkage planning [62,63]. Prominent examples emanate from Germany, where numerous cities such as Leipzig, Halle, and Dresden have adopted strategies to adapt to urban shrinkage, underpinned by national financial support. These strategies are predicated on the fiscal transfers and aid spanning regional levels in high-welfare nations [63,64]. In contrast to "SS", scholars within the European narrative framework tend to characterize this series of measures as "urban resurgence/revitalization/regeneration". However, the associated terminologies are seldom reflected in the titles, abstracts, and keywords of academic papers, thereby augmenting the difficulty of retrieving literature pertinent to these policies. We must concede this is a limitation of our research, yet the dispersion and lack of uniformity of these descriptive terms also curtail the international exchange and diffusion of numerous policy innovations. For instance, in China, the term "urban shrinkage" often evokes the concept of "smart decline" among many Chinese scholars. As articulated by Wiechmann, T. there is an urgent need for a transatlantic debate on this matter [3,65].

### 4.2. Main Findings of Bibliometric Statistics on SG and SS

The study provides the most comprehensive multidimensional quantitative comparative analysis of literature on SG and SS from 1997 to 2022. These results can visually demonstrate the developmental trajectory, current status, research hotspots, and trends in the study of SG and SS. The main findings are as follows: (1) Research on SG and SS emerged successively in the late 20th century and early 21st century. The publication volume related to SG was more than ten times that of SS (Table 1), but the research popularity of SS rapidly increased around 2010 (Figures 2 and 6). (2) Both SG and SS studies are primarily focused on environmental science and social science, with the former being more focused on environmental science and the latter being more oriented towards social science (Figures 3 and 7). (3) The study areas covered by SG are significantly greater and more extensive than those of SS. In addition to developed regions such as the United States and Europe, research on SG also encompasses many underdeveloped economic areas in Asia, Latin America, and Africa (Figure 8).

Our findings point to a seemingly apparent observation that SS, as a relatively new concept signaling good intentions to address urban shrinkage, is still on a path of consolidation (Figures 2, 3b, 4b, 6 and 8b). This is in stark contrast to the maturity of SG as an established terminology with well-defined guidelines widely recognized among scientists [43,44], professionals (e.g., Smart Growth Network), and governments (e.g., Environmental Protection Agency, United States). For SG, its discourse framing to mobilize stakeholders has shifted from growth versus no-growth (i.e., growth control) to "smart" versus "dumb" growth (i.e.,

sprawl and related problems) (Figure 5), reflecting a mindset change from mitigation to adaptation. Contrastingly, the discourse framing of SS—although intended to transform the traditional growth-oriented mindset of urban planning [19,52,66]—has been largely dominated by growth-oriented mitigation thinking (e.g., "urban development", "development project", and "redevelopment" in Figures 4b and 6), with adaptation thinking emerging only in the past five years (e.g., "degrowth coalition", "right sizing", and "small town" in Figure 6). In comparison with the broader studies on shrinking cities [18], especially since the early 2000s when the German architect Philipp Oswaltand his colleagues popularized the term "shrinking cities" [11], the scholarly interest in "smartly" addressing urban shrinkage and consensus on SS principles are remarkably and regretfully limited.

Although the research domain of SS is still at a nascent stage of conceptual and theoretical development, an encouraging finding of our study is that SS publications have been rapidly growing since 2010 (Figure 2a). This abnormal surge was likely attributable to the funding and implementation of the 7 EU FP project "Shrink Smart—The Governance of Shrinkage within a European Context" (2009–2012; no. 225193) [67]. Such a catalyzing effect of innovative projects, pioneering conferences, and consequent scholar communities on nurturing and advancing new ideas and research fields has been common, e.g., the establishment of the North American School of Landscape Ecology [68], the burgeoning of land system science [69], the birth of sustainability science [70], and the popularization of the concept of shrinking cities [11]. In this regard, funding agencies' bold financial support of such potentially transformative scholarly activities plays a decisive role. Such funding is particularly needed for synthesizing and promoting the existing "smartness" of addressing urban shrinkage in various places across the world; note that the "smart" wisdom of European and China's shrinking cities remains underrepresented [20,21,71] (cf. Figure 8).

In contrast to SS not gaining significant capacity in mobilizing research efforts for advancing "smart" urbanization, the term SG seems to have been losing its mobilizing capacity since 2013 (Figure 2a). As one of the reviewers proposed, this might result from a saturation of research topics, shifts in funding priorities, or changes in academic trends. Actually, in the past decade, there have been some criticisms of the compact urban development advocated by SG [72], including the inequality in urban land [73] and the decrease in residential satisfaction caused by high-density housing [74]. With the increasing influence of the people-centered research paradigm of sustainability science and the continuous expansion of embedded issues, more and more studies are re-examining the compact development advocated by SG from the perspective of the coordinated coupling of humans and the environment [75]. Although the number of new issues related to SG has gradually decreased after 2013, topics related to sustainability science (such as environmental sustainability, complexity, land cover, carbon dioxide, and green infrastructure; see Figure 6) have taken up a high proportion, indicating that SG is shifting towards a paradigm more in line with sustainability science. In fact, as early as 2012, Chapin [14] noticed this trend of shifting from SG to sustainable growth and called for the integration of issues such as industrial decline, employment, environment, and climate into future growth management.

### 4.3. Differences and Similarities between SG and SS

Undoubtedly, there are obvious differences between SG and SS. The former focuses more on spatial compactness, while the latter emphasizes scale simplification. Specifically, the "smartnesses" aspect of SG lies in the compact development of urban elements such as population, land, and industry. On one hand, SG strives to reduce urban sprawl by establishing organic connections between public transportation and land use, designing mixed-use communities, and enhancing the reuse of abandoned land within cities [48,76]. On the other hand, by implementing measures like Urban Growth Boundaries (UGBs), SG aims to maintain compact land use, allocate new urban land demand to existing built-up areas as much as possible, and minimize the impact on agriculture and ecology [40,77].

Unlike SG, the "smartnesses" aspect of SS lies in the intensive development of urban elements. SS mainly focuses on centralizing urban elements such as population, land, and industry to maintain the healthy operation of concentrated areas, especially in the context of a continuous decrease in population [53,65]. The specific approaches taken by different cities generally vary. Currently, the main approach is based on reducing the scale of land development, scientifically and reasonably controlling the size of the city, and strategically arranging the population based on the advantages and functional positioning of urban development. This transformation aims to change the traditional notion of necessary urban growth and maintain the intrinsic drive for sustainable urban development [56,65].

However, the differences mentioned above do not obscure the intrinsic connections between SG and SS. We find that although SG and SS have long been separate in research and practice, there are at least three similarities between them.

First, they share similar value concepts. Due to historical factors like deindustrialization and suburbanization, urban development challenges such as inner-city population decline, industrial stagnation, and low land use efficiency have emerged one after another [78]. Both SG and SS were proposed to address specific problems arising from the urbanization process, and both are committed to sustainable urban development [36,77].

Second, they have similar core objectives. Whether addressing urban sprawl or urban shrinkage, the core goal of both SG and SS is to achieve optimal allocation of urban elements such as population, land, and industry [79,80], resolving issues of unsustainability caused by urban element misallocation (spatial misallocation, scale misallocation, etc.).

Lastly, they share similarities in disciplinary paradigms and research themes. Research related to SG and SS shows an interdisciplinary characteristic, merging social sciences and environmental sciences. Both seek to improve the physical form and landscape pattern of cities [81,82], while also considering policy-making and management issues [40]. For example, they emphasize the collaborative participation of diverse stakeholders like government, market forces, residents, and social groups, and direct research interests toward discussions of social equity [83,84].

### 4.4. Implications and Future Directions

Currently, both SG and SS fall short in their respective commitment to "urban sustainability" [85], and there is a need to shift the fragmented discourse on SG and SS. Building on the analysis of the similarities in and differences between SG and SS in the previous sections, we abstracted the core value concepts from the numerous principles and initiatives of SG and SS, and attempted to construct a cohesive theoretical framework of "Smart Development" (Figure 9) that can comprehensively consider both SG and SS. This framework, starting from the issues of urban shrinkage and urban sprawl arising from the urbanization process, reexamines the concepts and practical methods of SG and SS from the three dimensions of social, economic, and environmental aspects of urban sustainability. It integrates the core concepts of SG and SS into "Smart Development" to achieve a more comprehensive and coordinated approach to urban sustainable development.

In terms of striving to enhance urban sustainability, SG and SS are not in opposition but rather can complement each other in a coordinated manner. The extensive planning practices of SG can provide experiential and methodological support for SS in shaping urban landscape patterns and optimizing and restructuring public service networks, thus offering valuable insights to avoid spatial fragmentation and systemic dislocation of public services in shrinking cities [39]. The discussions in SS on development rights, equity issues, and unique insights into urban development trends can enrich the value connotations of SG and provide inspiration on how to truly improve human well-being [57]. We hope to achieve the coupling and coordination of SG and SS based on this theoretical complementarity, merging the two into "Smart Development" to comprehensively consider the three dimensions of urban sustainability and promote sustainable urban development.

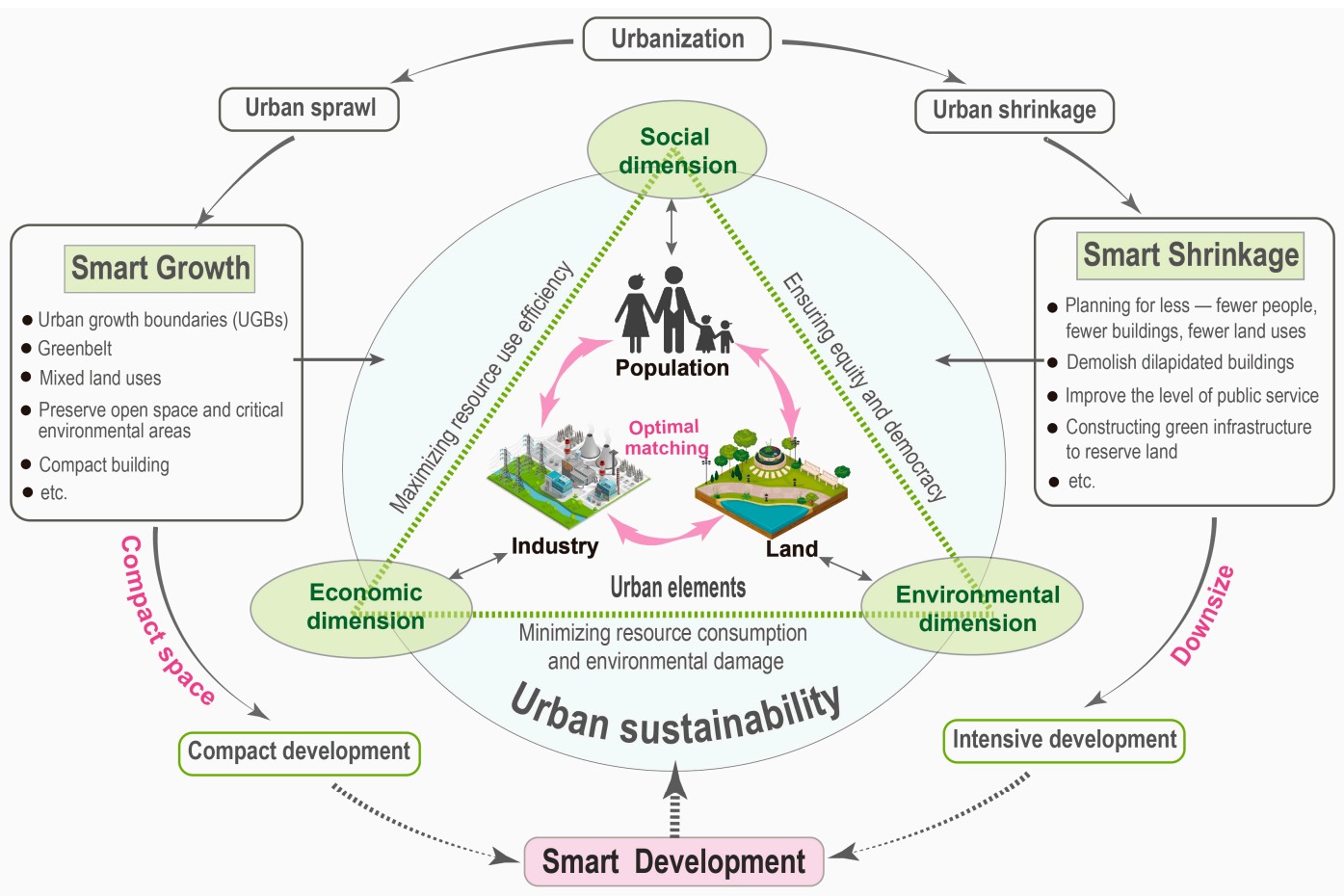

**Figure 9.** Theoretical framework of "Smart Development" for urban sustainable development.

At the theoretical level, it is necessary to change the existing situation of treating urban growth and urban shrinkage in isolation, and consider the issues of urban growth and shrinkage from a more macro and holistic perspective. Specifically, it includes simultaneously measuring and assessing urban growth and shrinkage, comprehensively analyzing the driving mechanisms of growth and shrinkage from a systemic perspective, and predicting future scenarios of urban growth and shrinkage, as well as potential challenges and opportunities based on current trends and possible influencing factors. It is encouraging that there is an increasing amount of research simultaneously measuring urban growth and shrinkage [86,87]. Moreover, with the advancement in artificial intelligence and machine learning technologies, it has become increasingly possible to more accurately predict various trends and scenarios for the future of cities [88–90].

At the practical level, efforts can be strengthened in the following aspects. Firstly, the selection of strategies must be based on scenarios that match the urban development, and a comprehensive action plan should be formulated according to the scientific prediction of the temporal and spatial dynamics of the urban population, so as to avoid the mismatch in the scale and space of urban elements. Secondly, it is essential to approach the coordination of interests among different groups with greater caution, establishing mechanisms for dialogue and interest communication, and involving a more diverse set of actors in responding to urban sprawl and urban shrinkage [91]. Finally, for the coordinated governance of urban sprawl and urban shrinkage, positive interactions across multiple levels are necessary. On one hand, emphasis should be placed on the coordination among various levels within the city, from individuals and families to neighborhoods, communities, functional areas, and the city. On the other hand, the city itself should be positioned within a larger urban system and network. In this way, the rational flow of resource elements can be ensured

among small towns, medium-sized cities, large cities, mega-cities, metropolitan areas, and even national and global cities, ultimately achieving urban sustainable development in which the overall functionality exceeds the sum of its parts.

## 5. Conclusions

Land-related approaches to advancing sustainability are increasingly recognized for their critical role in shaping the future of urban environments. This paper explores the intertwined roles of SG and SS as complementary strategies within this context, emphasizing their importance in guiding urbanization towards sustainability. By employing bibliometric analysis, we systematically explored the evolution and thematic focus of SG and SS research, highlighting the distinct yet interconnected pathways they offer for sustainable urban development. Our findings reveal a divergence in the scope and clarity of research themes between SG and SS, with SG commanding a broader and more defined research landscape. Despite this, SS has experienced a notable surge in interest since 2010, signaling a shift in the focus of sustainability discussions towards embracing urban shrinkage as a viable and necessary approach to sustainability in post-industrial contexts. This shift underscores the growing recognition of the need to balance urban expansion with shrinkage, reflecting a holistic view of urban dynamics that incorporates both growth and shrinkage as integral to sustainable development.

SG, rooted in New Urbanism, emphasizes the optimization of urban physical form and the mitigation of environmental impacts, thereby addressing the spatial and ecological aspects of sustainability. On the other hand, SS, emerging from critical reflections on urban decline, prioritizes economic revitalization and the enhancement of resident well-being, thus tackling the economic and social dimensions of sustainable urban living. These dual focuses underscore the necessity of a balanced approach that considers both environmental integrity and human well-being. The integration of SG and SS into a unified framework for urban sustainability is crucial for navigating the complexities of urban development. The unified theoretical framework of "Smart Development" (Figure 9) proposed in this paper organically integrates the core concepts of SG and SS, and can promote more comprehensive and coordinated urban sustainable development. As cities evolve into more dynamic and complex systems, the recognition of SG and SS as normal processes of urbanization highlights the need for adaptable and comprehensive strategies that address the full spectrum of sustainability challenges. By fostering more research on SS across various contexts and integrating it with SG principles, we can better understand and implement land-related approaches that advance sustainability in an increasingly urbanized world. This holistic approach not only addresses the immediate challenges of urban growth and decline but also lays the groundwork for a sustainable and resilient urban future.

**Author Contributions:** Conceptualization, Y.Y.; methodology, Z.D. and B.-B.Z.; software, Z.D. and B.-B.Z.; validation, B.-B.Z., Y.Y. and Y.L.; formal analysis, Z.D.; investigation, Z.D.; resources, Y.Y.; data curation, Z.D. and B.-B.Z.; writing—original draft preparation, Z.D. and Y.Y.; writing—review and editing, Y.Y., B.-B.Z. and Y.L.; visualization, Z.D. and B.-B.Z.; supervision, B.-B.Z. and Y.Y.; project administration, Y.Y.; funding acquisition, Y.Y. All authors have read and agreed to the published version of the manuscript.

**Funding:** This work was supported by the National Social Science Fund of China (19CJL025).

**Data Availability Statement:** Data will be available upon reasonable request.

**Conflicts of Interest:** The authors declare no conflicts of interest.

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
