# Peer review of "Smart Growth and Smart Shrinkage: A Comparative Review for Advancing Urban Sustainability"

_land, doi:10.3390/land13050660_

Round 1
Reviewer 1 Report
Comments and Suggestions for Authors
This study bridges the gap in the fragmented discourse surrounding smart growth and smart shrinkage by conducting a comprehensive comparative review on the respective literatures. The authors did an excellent job. The paper is recommended for publication after revision.
1. In the Introduction section, the meaning, origin and context of "smart growth" and "smart shrinkage" need to be explained in detail so that readers can first understand these two concepts.
2. What is the intrinsic link between "smart growth" and "smart shrinkage"? The authors need to elaborate further in the discussion or conclusion section.
3. How can "smart growth" and "smart shrinkage" be integrated into a unified theoretical framework? What does this theoretical framework look like? It is suggested that the authors construct a theoretical framework to integrate "smart growth" and "smart shrinkage ".
4. The last column in table 1, "Ratio" requires explanation.
5. It is suggested that Figure S1 could be inserted into the manuscript.
6. In line 356, "smart growth" should be replaced by "smart decline" or "smart shrinkage".
Reviewer 2 Report
Comments and Suggestions for Authors
Overall, this paper presents a valuable comparative review of Smart Growth and Smart Shrinkage research, offering insights into their trajectories, thematic coverage, and implications for urban sustainability. However, some adjustments are required to develop the content before publication. My comments for development are as follows:
Abstract
The abstract mentions "notable shift in research focus" and "surging interest in SS around 2010", but it could benefit from more specific details about these shifts. Providing specific percentages or trends could enhance the impact of the review.
introduction
While the introduction outlines the general objective of bridging the gap between Smart Growth (SG) and Smart Shrinkage (SS) research, it could benefit from more specific statements regarding the intended outcomes or contributions of the study. Clarifying the specific research questions or hypotheses being addressed would provide readers with a clearer understanding of the study's purpose.
In addition, the introduction could be strengthened by more effectively integrating relevant theoretical frameworks or prior research findings to provide a stronger theoretical foundation for the study.
Materials and Methods section
The section provides information on the number of papers sampled for SG and SS, as well as their bibliographic details. However, it could be helpful to include a brief discussion on how the sample size was determined and whether any criteria were applied for inclusion or exclusion of papers.
While the section mentions the use of the Mann-Kendall test and Pettitt's test to assess trends and changes in publication and citation frequencies, it could benefit from providing a brief explanation of these tests for readers who may not be familiar with them. Including a sentence or two to describe the purpose and interpretation of these tests would enhance clarity.
While the section mentions the use of the Mann-Kendall test and Pettitt's test to assess trends and changes in publication and citation frequencies, it could benefit from providing a brief explanation of these tests for readers who may not be familiar with them. Including a sentence or two to describe the purpose and interpretation of these tests would enhance clarity.
Results
Page 4, The result section notes a rapid growth in SG-related research following its emergence in 1997, with a peak in publication rates around 2013. This pattern suggests a significant interest and investment in SG concepts within the academic community. However, the subsequent decline in publication rates raises questions about the sustainability of this interest. It would be beneficial to explore potential reasons behind this decline, such as saturation of research topics, shifts in funding priorities, or changes in academic trends.
The observation that SS-related research began in 2002 and experienced a period of tepid growth until 2010, followed by a significant surge in research intensity, is intriguing. It prompts questions about the factors that contributed to the delayed uptake of SS concepts and the subsequent rapid expansion. Exploring potential catalysts for this sudden surge, such as shifts in policy priorities, emerging socio-economic challenges, or advancements in research methodologies, could provide valuable insights into the evolution of SS research.
The observation that SS hot keywords focus on the conceptual construction and theoretical research of urban shrinkage underscores the developmental stage of the SS domain. In contrast, the prevalence of classical theories and practical urban planning topics in SG hot keywords suggests a more mature and established research agenda. Exploring the implications of these differences in conceptual development for the advancement of urban sustainability discourse can inform strategies for bridging theoretical and practical perspectives in SG and SS research.
Discussion
The discussion section provides a nuanced comparison between Smart Growth (SG) and Smart Shrinkage (SS), highlighting their similarities, differences, research progress, and challenges. However, some points should be developed in more depth:
1. Distinct Trajectories and Thematic Coverage
2. Research Progress and Maturity
3. Challenges and Future Directions
Conclusion:
The conclusion effectively summarizes the key findings and implications of the study. However, it could be strengthened by providing more specific recommendations for future research or practical applications based on the findings. Additionally, explicitly addressing the limitations of the study and avoiding overgeneralization of the findings would improve the clarity and impact of the conclusion.
Reviewer 3 Report
Comments and Suggestions for Authors
Please see attached file

Minor editing
Round 2
Reviewer 2 Report
Comments and Suggestions for Authors
Thank you for answering the comments. The manuscript has been developed in a good way. Well done.